# The Role of Bitter Melon in Breast and Gynecological Cancer Prevention and Therapy

**DOI:** 10.3390/ijms24108918

**Published:** 2023-05-17

**Authors:** Iason Psilopatis, Kleio Vrettou, Constantinos Giaginis, Stamatios Theocharis

**Affiliations:** 1First Department of Pathology, Medical School, National and Kapodistrian University of Athens, 11527 Athens, Greece; 2Department of Food Science and Nutrition, School of Environment, University of Aegean, 81400 Lemnos, Greece

**Keywords:** bitter melon, bitter gourd, momordica charantia, breast cancer, ovarian cancer, cervical cancer, endometrial cancer, uterine cancer, vaginal cancer, vulvar cancer

## Abstract

Phytotherapy has long represented a widely accepted treatment alternative to conventional therapy. Bitter melon is a vine with potent antitumor effects against numerous cancer entities. To date, no review article has, however, been published on the role of bitter melon in breast and gynecological cancer prevention and therapy. The current work constitutes the most comprehensive, up-to-date review of the literature, which highlights the promising anticancer effects of bitter melon on breast, ovarian, and cervical cancer cells and discusses future research recommendations.

## 1. Introduction

In the Western world, standard medical care represents the widely accepted, proper treatment for certain types of disease and is based on conventional medicine, a system in which healthcare professionals treat medical conditions using drugs, radiation, and/or surgery [1]. In Eastern cultures, on the contrary, complementary or alternative medicine is still very popular and incorporates various botanicals and nutritional products, ranging from herbal and dietary supplements to vitamins [2]. In fact, according to the World Health Organization (WHO), around 80% of the world’s population relies on traditional systems of medicine for primary healthcare needs, with the Indian and Chinese traditional medicinal systems constituting the major and oldest ones [3]. Interestingly, plant products, either whole extract or bioactive components, along with fruits and vegetables, may even contribute to the reduction in cancer risk, thanks to the cumulative effect of diverse bioactive phytochemicals, minerals, vitamins, fibers, and proteins [4].

Bitter melon, bitter gourd, karela, or *Momordica charantia*, is a vine belonging to the Cucurbitaceae family, which is widely cultivated in Asian, African, and South American countries [5]. More precisely, it is a rich source of phytochemicals and has the highest nutritional value among cucurbits [6]. Its biological activity may be mainly attributed to its major chemical constituents: cucurbitane-type triterpenoids, cucurbitane-type triterpene glycosides and their aglycones, flavonoids, phenolic acids, fatty acids, essential oils, lectins, amino acids, goyasaponins, sterols, as well as several proteins [5,7]. The proportion of these chemical constituents varies, depending on the different varieties of bitter melon, the different origins and cultivation conditions, or the harvest times [4]. Bitter melon extract and its active ingredients have been extensively studied in diverse cell line-based and animal models and reported to exhibit promising effects on the chemoprevention and therapy of skin, brain, oral, lung, liver, colon, stomach, blood, prostate, renal, and pancreatic cancers [4]. Notably, the molecular mechanisms of cancer prevention and therapy do not seem to differ among the different cancer entities, with bitter melon enhancing Reactive Oxygen Species (ROS) production, modulating the cell membrane, inducing apoptosis and autophagy, causing epigenetic modifications, as well as interacting with the DNA, the RNA, or numerous proteins [4,6].

Ovarian, corpus uteri, cervical, vaginal, and vulvar malignant tumors represent the main types of gynecological cancers [8]. Between 2012 and 2016, about 94,000 women were diagnosed with gynecologic cancer annually, with uterine cancer representing the most and common cancer entity and vaginal cancer representing the least common gynecological cancer entity [9]. Apart from skin cancers, breast cancer is considered the most common cancer in women in the United States, with an estimated annual incidence of 297,790 new cases and mortality exceeding 43,700 for 2023 [10]. Despite significant improvements in both breast/gynecological cancer prevention and therapy over the past few years, such statistics render imperative the identification of further innovative agents for the effective management of breast and gynecological cancer patients. The aim of this review is to closely investigate the role of bitter melon as a natural compound in breast and gynecological cancer prevention and therapy.

## 2. The Preventive and Therapeutic Potential of Bitter Melon in Selected Cancer Entities

Over the past three years, numerous study groups have closely investigated the preventive and/or therapeutic potential of bitter melon in certain cancer entities. Sur et al. reported on the diverse effects of the crude extract and some of the isolated pure compounds of bitter gourd on oral squamous cell carcinoma. More precisely, cell cycle modulation, modulation in cell signaling, induction of apoptotic cell death, inhibition of glycolysis and lipid metabolism, as well as induction of anticancer immunity, represent only a few of the underlying molecular mechanisms of bitter melon in the prevention of oral squamous cell carcinoma [11]. In the context of pancreatic cancer, Kandhari et al. compared the pre-clinical efficacy of the lyophilized-juice and aqueous-methanolic extracts of the Chinese and the Indian *Momordica charantia* variants, respectively, and concluded that both cultivars exhibit comparable effectiveness against pancreatic cancer growth and progression. Both bitter melon juice and bitter melon extract downregulated cell proliferation and tumor growth, induced apoptotic cell death, inhibited capillary tube formation, as well as hindered angiogenesis in pancreatic cancer tumor xenograft models [12]. As far as head and neck cancer is concerned, the bitter gourd bioactive secondary metabolite momordicine-I reduced human head and neck cancer cell viability in a dose-dependent way, with negligible side effects on non-cancerous cells, constraining c-Met and its downstream signaling molecules via the inhibition of Signal Transducer and Activator of Transcription 3 (STAT3) in head and neck cancer cells, and it reduced head and neck cancer tumor growth in nude mice xenograft models, involving c-Met and downstream signaling [13]. Furthermore, Zhou et al. assessed the potential of the ribosome-inactivating protein MAP30 isolated from bitter gourd and underlined a reduced survival and proliferation of human liver cancer cells in a dose-dependent manner following treatment with MAP30. More accurately, the application of bitter melon resulted in the induction of cell cycle arrest in G0/G1 phase, an increase in the level of ROS, the downregulation of Poly (ADP-Ribose) Polymerase (PARP)-1 and caspase-3, alongside the upregulation of cleaved caspase-9 [14]. Last, but not least, Zhang et al. outlined that MAP30 inhibits bladder cancer cell migration and invasion in vitro via the suppression of the Akt pathway and the epithelial-mesenchymal transition process [15]. Taken altogether, bitter melon seems to represent a vine with potent antitumor effects against different cancer entities.

## 3. The Role of Bitter Melon in Breast Cancer Prevention and Therapy

The literature review was conducted using the MEDLINE and LIVIVO databases. By employing the search terms “bitter melon” and “breast cancer”, we were able to identify a total of 19 relevant original research articles published between 1994 and 2023.

In 1994, Rybak et al. were the first to identify breast cancer cell lines as one of the most sensitive cancer cell lines in relation to the Momordica Anti-HIV Protein 30 kDa (MAP30) and Gelonium Anti-HIV Protein 31 kDa (GAP31) bitter melon-isolated proteins treatment [16]. Huang et al. measured the anticancer capacities of MAP30 and GAP31 by proliferation of MDA-MB-231 breast cancer cells and concluded that their endopeptidase digestion generates fragments with a potent anticancer activity, whereas their antitumor effects do not rely on the ribosome inactivation activity [17]. Accordingly, Lee-Huang et al. examined the efficacy of MAP30 and GAP31 on MDA-MB-231 cells and reported reduced in vitro cancer cell proliferation and Human Epidermal growth factor Receptor 2 (HER2) gene expression, alongside significantly higher survival rates in treated human breast cancer bearing Severe Combined ImmunoDeficiency (SCID) xenograft mouse models [18].

Bai et al. isolated the triterpenoid 3β, 7β, 25-trihydroxycucurbita-5,23(E)-dien-19-al (TCD) from wild bitter melon and showed that TCD inhibited the proliferation of MDA-MB-231 and MCF-7 breast cancer cells, as well as provoked breast cancer cell apoptosis. More precisely, the aforementioned triterpenoid repressed Akt-Nuclear Factor kappa B (NF-κB) signaling, activated both the p38 mitogen-activated protein kinase and p53, generated ROS, downregulated histone deacetylase protein expression, as well as promoted cytoprotective autophagy [19]. In 2013, Weng et al. noted that the triterpene 3β,7β-dihydroxy-25-methoxycucurbita-5,23-diene-19-al (DMC) could effectively induce apoptosis in MCF-7 and MDA-MB-231 cells by activating the Peroxisome Proliferator-Activated Receptor (PPAR) γ. On the one hand, DMC inhibited mammalian Target Of Rapamycin (mTOR)-p70S6K signaling via Akt repression and 5′ AMP-activated Protein Kinase (AMPK) upregulation and blocked the expression of cyclin D1, Cyclin Dependent Kinase 6 (CDK6), B-cell lymphoma 2 (Bcl-2), X-linked Inhibitor of Apoptosis Protein (XIAP), cyclooxygenase-2, NF-κB, and estrogen receptor α. On the other hand, DMC successfully induced Growth Arrest- and DNA Damage-Inducible gene 153 (GADD153) and Glucose-Regulated Protein 78 (GRP78) expression, alongside a cytoprotective autophagy [20]. Some years later, the same study group proposed the triterpenoid 5β,19-epoxy-19-methoxycucurbita-6,23-dien-3β,25-diol as a potent PPARγ activator in MCF-7 breast cancer cells, which diminished the expression of histone deacetylase 1, increased the phosphorylation of p53, generated ROS, and promoted G1 cell cycle arrest through cyclin D1 and CDK6 downregulation [21].

In 2019, Lepionka et al. fed Sprague-Dawley rats experimental diets supplemented with *Momordica charantia* extract and outlined that, even though this diet modification boosted *cis*-9, *trans*-11 conjugated linoleic acid levels, the breast cancer incidence did not drop in rats treated with 7,12-dimethylbenz[a]anthracene. Furthermore, a profound impact on serum fatty acids content was noted, while co-existing cancerous process contributed to the reduction in saturated fatty acids, monounsaturated fatty acids, polyunsaturated fatty acids, and 8-iso prostaglandin F2α, in serum [22]. Two years later, Białek et al. evaluated the effect of bitter gourd extract on the lipidomic profile of cardiac tissue in female Sprague-Dawley rats with breast cancer. This dietary supplementation resulted in the diminution of cholesterol levels via the downregulation of the endogenous conversion of squalene to cholesterol in the heart tissue. Furthermore, the cardiac incorporation of conjugated fatty acids was evidently less in the cancerous process, whereas malondialdehyde levels experienced only slight changes [23].

Cao et al. investigated the effects of the bitter melon-extracted ribosome inactivating protein alpha-momorcharin on the inhibition of human breast cancer by purifying it by the means of column chromatography and, consecutively, injecting it into a xenograft nude mouse model induced by MDA-MB-231 and MCF-7 breast cancer cells. Alpha-momorcharin was found to efficiently inhibit in vivo tumor growth, increase caspase-3 activities, as well as lead to cell cycle arrest at the G0/G1 or G2/M phases [24]. Similarly, Deng et al. explored the anticancer activity of alpha-momorcharin in EMT-6 and MDA-MB-231 transplanted tumor mouse models and demonstrated that polyethylene glycolylation increased its plasma half-life in vivo. Interestingly, modification of alpha-momorcharin with polyethylene glycol correlated with an enhanced anticancer effectiveness, alongside a more moderate toxic profile [25]. Besides, Fang et al. purified the 14-kDa ribonuclease MC2 in the seeds of *Momordica charantia*, which exhibited not only cytostatic, but also cytotoxic effects on MCF-7 breast cancer cells via karyorrhexis, chromatin condensation, DNA fragmentation, caspase-7/-8/-9 activation, B-Cell Lymphoma 2 (BCL-2) Antagonist/Killer (BAK), and cleaved PARP production, as well as p38, c-Jun N-terminal Kinase (JNK), Extracellular signal-Regulated Kinase (ERK), and Akt differential activation [26].

Ehigie et al. treated MDA-MB-436 breast cancer cells with different fractions derived from the aqueous extract of the leaves of bitter melon, which downregulated mitochondrial membrane potential and intracellular ATP levels, but enhanced ROS levels. These cytotoxic effects of bitter melon were mediated by loss of mitochondrial function via loss of respiration, leading to cell death, rather than by the classical release of cytochrome c or caspase-3 activated apoptosis [27]. Moreover, Grossmann et al. focused their research on the effects of eleostearic acid, a component of bitter melon seed oil, on MDA-MB-231 and MDA-ERα7 human breast cancer cells. Eleostearic acid hindered cancer cell proliferation, induced apoptosis, caused mitochondrial membrane potential loss, provoked apoptosis-inducing factor and endonuclease G nuclear translocation, as well as arrested the cancer cell cycle. Nevertheless, lipid peroxidation was shown to significantly determine the inhibitory actions of eleostearic acid [28]. Muhammad et al. observed that bitter gourd extract application to breast cancer cells led to the induction of autophagosome-bound Long Chain 3 (LC3)-B, the accumulation of p62/SQSTM1 (p62) protein, an enhanced phospho-AMPK expression, as well as the downregulation of the mTOR/Akt signaling pathway. In vivo, the aforementioned extract inhibited tumor growth by upregulating p62 accumulation and provoking autophagy and apoptosis in syngeneic and xenograft mouse models [29]. After treating MCF-7 and MDA-MB-231 breast cancer cells with *Momordica charantia* extract, Ray et al. also described a significant reduction in cancer cell proliferation, accompanied by an amplified PARP cleavage and caspase activation. Survivin and claspin expression levels remained low, MCF-7 cells accumulated during the G2/M cell cycle phase, p53, p21, and pChk1/2, were upregulated, whereas cyclins B1 and D1 expression was hindered [30]. Nagasawa et al. monitored the effects of bitter gourd extract on spontaneous breast cancer development in SHN virgin mice, which significantly inhibited in vivo tumor growth with negligible side effects [31].

Feng et al. proved that even 4T1 triple-negative breast cancer cells might successfully internalize bitter gourd-derived vesicle extracts, which inhibited cancer cell proliferation and migration, while it also stimulated ROS production and disrupted mitochondrial function. Of note, bitter melon-derived vesicle extracts also dramatically diminished breast cancer growth in female BALB/c mice with negligible side effects [32]. Similarly, Shim et al. applied bitter melon extract to triple-negative breast cancer cell lines and measured low esterified cholesterol, Acetyl-CoA Acetyltransferase 1 (ACAT-1), sterol regulatory element-binding proteins-1 and -2, fatty acid synthase, and low-density lipoprotein receptor expression levels. Remarkably, this extract also inhibited both tumor growth and ACAT-1 expression in triple-negative breast cancer xenograft mouse models [33].

Last, but not least, Kilcar et al. tested the effects of the bitter gourd extract on the uptake of Technetium-99m-labeled paclitaxel against MCF-7 and MDA-MB-231 breast cancer cells and highlighted a significant estrogen receptor-dependent interaction [34].

Altogether, these results indicate that bitter melon and its bioactive components may successfully inhibit breast cancer development and progression both in vitro and in vivo.

Table 1 briefly summarizes the aforementioned findings.

## 4. The Role of Bitter Melon in Ovarian Cancer Prevention and Therapy

The search of the MEDLINE and LIVIVO databases using the terms “bitter melon” and “ovarian cancer” revealed a total of four relevant original research articles published between 2016 and 2020.

Martin et al. proved that bitter taste receptors are prevalent in human epithelial ovarian cancer cell lines and speculated that noscapine stimulation by natural bitter compounds, such as the bitter gourd extract, might induce ovarian cancer cell apoptosis in a receptor-dependent manner [35].

Chan et al. isolated the protein MAP30 from bitter gourd seeds and showed that MAP30 was therapeutic against ovarian cancer cells. More precisely, the aforementioned bioactive protein prevented cancer cell migration, invasion, and proliferation, in OVCA433 and ES2 ovarian cancer cells, while co-administration with cisplatin resulted in clearly enhanced cytotoxic effects. Moreover, MAP30 activated AMPK signaling, provoked cell cycle arrest in the S-phase, suppressed GLUcose Transporter (GLUT)-1/-3-mediated glucose uptake, lipid droplet formation, and adipogenesis, as well as triggered ROS-mediated apoptosis and ferroptosis. In vivo, MAP30 and cisplatin synergistically repressed ovarian cancer dissemination and growth without significant side effects on liver and kidney functions in an ovarian cancer ascites mouse model [36]. Pitchakarn et al. examined the combinational effects of the bitter melon vine-isolated chemical compound kuguacin J and cisplatin/paclitaxel in drug-resistant SKOV3 ovarian cancer cells. Kuguacin J augmented SKOV3 sensitivity to cisplatin/paclitaxel by diminishing survivin levels and inducing cleavage of PARP and caspase-3 [37]. Additionally, Yung et al. co-treated ovarian cancer cells with *Momordica charantia* extract and cisplatin, which attenuated tumor growth both in vitro and in vivo in a mouse xenograft model with negligible side effects. Concerning the molecular background, the bitter melon extract seemed to activate AMPK in a Ca^2+^/calmodulin-dependent protein kinase-dependent way and to suppress the mTOR/p70S6K and/or the AKT/ERK/Forkhead Box M1 (FOXM1) signaling pathway. Remarkably, the anticancer effects of bitter melon varied within its diverse varieties, implying that the amount of its anticancer constituents may vary [38].

All in all, bitter melon, alone or in combination with standard chemotherapy, exerts potent antitumor effects in ovarian cancer.

Table 2 briefly summarizes the aforementioned findings.

## 5. The Role of Bitter Melon in Uterine Cancer Prevention and Therapy

By employing the search terms “bitter melon”, “uterine cancer” and “endometrial cancer”, we were unable to identify any relevant original research articles.

## 6. The Role of Bitter Melon in Cervical Cancer Prevention and Therapy

The literature search of the MEDLINE and LIVIVO databases for the search terms “bitter melon” and “cervical cancer” led to the identification of a total of three relevant original research articles published between 2003 and 2012.

Limtrakul et al. obtained chemical modulators from bitter gourd extracts in order to investigate their abilities to regulate the function of the 170 kDa P-glycoprotein and the multidrug resistance phenotype in the KB-V1 human cervical cancer cells. Interestingly, only the leaf extracts seemed to have a positive effect on the intracellular accumulation of [^3^H]-vinblastine and to sensitize KB-V1 cells to vinblastine [39]. Furthermore, Pitchakarn et al. isolated kuguacin J from bitter melon leaves and reported that this active component sensitized KB-V1 cells to vinblastine and paclitaxel by upregulating the intracellular accumulation of rhodamine123, calcein AM, and [^3^H]-vinblastine, as well as blocking the incorporation of [^125^I]-iodoarylazidoprazosin into P-glycoprotein in a concentration-dependent manner [40]. Last, but not least, Pongnikorn et al. investigated the effects of *Momordica charantia* application on cervical cancer patients undergoing radiotherapy and underlined that bitter gourd ingestion might not affect natural killer cell count, but seemingly had a negative impact on P-glycoprotein level on natural killer cell membranes [41].

Taken altogether, bitter gourd might effectively improve the therapeutic efficacy of standard chemoradiotherapy in cervical cancer.

Table 3 briefly summarizes the aforementioned findings.

## 7. The Role of Bitter Melon in Vaginal Cancer Prevention and Therapy

No results were found for the search terms “bitter melon” and “vaginal cancer” in neither the MEDLINE nor the LIVIVO databases.

## 8. The Role of Bitter Melon in Vulvar Cancer Prevention and Therapy

By employing the search terms “bitter melon” and “vulvar cancer”, we were not able to find any relevant original research articles.

## 9. Discussion

Gynecologic and breast cancer patients often resort to complementary medicine in addition to conventional Western therapy [42,43]. This tendency may be mainly attributed to the fact that conventional chemoradiotherapy is usually associated with adverse effects that negatively influence the quality of life and discourage the affected women [44]. As a consequence, many patients find hope in natural therapeutic alternatives, which exhibit promising treatment outcomes with less side effects than standard therapy [45]. More precisely, a great number of review articles has, to date, been published on the efficacy and safety of phytotherapy on the prevention and treatment of gynecologic and/or breast cancer [46,47,48,49]. Nonetheless, no study review has been published on the concrete role of bitter melon in breast and gynecological cancer prevention and therapy. The current work, to our knowledge, constitutes the most inclusive, up-to-date review of the literature that comprehensively summarizes the numerous effects of *Momordica charantia* on breast and gynecological cancer cells (Figure 1).

With a total of 19 relevant original research articles published between 1994 and 2023, breast cancer undoubtedly represents the most studied cancer entity. Most study groups have employed the human breast cancer cell lines MCF-7 and MDA-MB-231 and discovered that the bitter melon extract might efficiently inhibit the in vitro progression of both estrogen receptor-positive and estrogen receptor-negative breast cancer cells. Of note, its bioactive components seem to interact with histone deacetylases and PPARs, which have been proposed to play a significant role in breast and gynecologic cancer development and progression [50,51,52,53,54,55]. In addition, bitter gourd extract enhanced the cytotoxic effects of standard chemotherapy on breast cancer cells, hence endorsing the assumption of potent synergistic antitumor effects. In vivo, bitter melon extract successfully inhibited tumor growth in (triple-negative) breast cancer mouse xenograft models with moderate toxic side effects and a profound impact on serum and cardiac fatty acids content.

In the context of ovarian cancer, karela extract prevented cancer cell migration, invasion, and proliferation, and it induced apoptotic cell death. In combination with cisplatin/paclitaxel, the aforementioned extract exerted promising synergistic therapeutic effects and even increased the sensitivity of chemoresistant ovarian cancer cells to standard chemotherapy.

Two research groups have, to date, investigated the effects of *Momordica charantia* application to KB-V1 cervical cancer cells, which both focused on its regulatory capacities on the P-glycoprotein and the multidrug resistance phenotype [39,40]. Most importantly, Pongnikorn et al. launched the first reported clinical trial and treated cervical cancer patients undergoing radiotherapy with bitter melon, thereby paving the way for further research in the clinical setting [41].

Unfortunately, no original research article has, to date, been published on the role of bitter melon in uterine, vaginal, or vulvar cancer.

It should be noted that most data have been derived by pre-clinical studies, highlighting the emergent need for performing well designed clinical studies in cancer patients. Up to now, diverse varieties of bitter melon with different amounts of its bioactive ingredients and different treatment periods and administration forms have been tested, also underlining the need to use more similar experimental protocols to obtain more comparable and reliable evidence on the anticancer effects of bitter melon against breast and gynecological cancer.

## 10. Conclusions

In summary, bitter melon seems to represent a promising natural agent which, alone or in combination with standard chemoradiotherapy, may help prevent and treat breast, ovarian, and cervical cancer. Further research is, nevertheless, needed, in order to confirm the reported in vitro and in vivo study results and to evaluate the feasibility of bitter gourd ingestion in clinical trials incorporating large patient collectives. Indeed, only large randomized-controlled trials would allow for safe, reproducible, and statistically significant results, which would not only prove the benefits of the clinical application of *Momordica charantia*, but also help recognize eventual unidentified side effects. Last, but not least, the promising results from the summarized studies, focusing on breast, ovarian, and cervical cancer, should motivate researchers to also examine the role of bitter melon in uterine, vaginal, and vulvar cancer prevention and therapy.

## Figures and Tables

**Figure 1 ijms-24-08918-f001:**
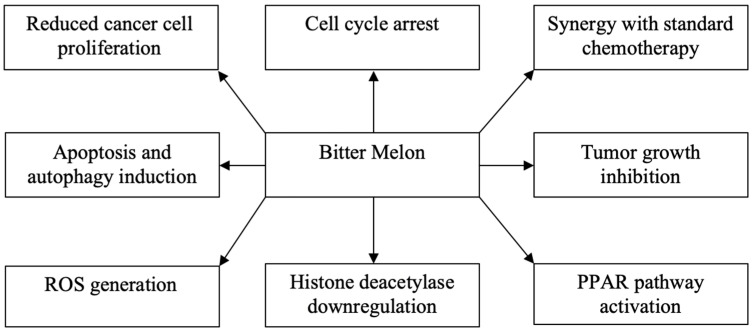
The anticancer effects of bitter melon in breast and gynecological cancer.

**Table 1 ijms-24-08918-t001:** The role of bitter melon in breast cancer prevention and therapy.

Study	Study Model	Main Findings
Rybak et al. [16]	BT20	Sensitivity to MAP30 and GAP31
Huang et al. [17]	MDA-MB-231	Dependency on endopeptidase digestion
Lee-Huang et al. [18]	MDA-MB-231	Reduced in vitro cancer cell proliferation and *HER2* expression Significantly higher survival rates in treated SCID mice
Bai et al. [19]	MCF-7 MDA-MB-231	Cancer cell proliferation inhibition Apoptosis and autophagy induction ROS generation Histone deacetylase downregulation
Weng et al. [20]	MCF-7 MDA-MB-231	Apoptosis and autophagy induction PPARγ pathway activation
Weng et al. [21]	MCF-7	PPARγ pathway activation ROS generation Histone deacetylase 1 downregulation Cell cycle arrest
Lepionka et al. [22]	Sprague-Dawley rats	Elevated *cis*-9, *trans*-11 conjugated linoleic acid levels Profound impact on serum fatty acids content
Białek et al. [23]	Sprague-Dawley rats	Diminution of cholesterol levels in the heart tissue Less cardiac incorporation of conjugated fatty acids in the cancerous process
Cao et al. [24]	MCF-7 MDA-MB-231 MDA-MB-453	In vivo tumor growth inhibition Enhanced caspase-3 activities Cell cycle arrest
Deng et al. [25]	MDA-MB-231 EMT-6	Enhanced anticancer effectiveness and more moderate toxic profile after polyethylene glycolylation
Fang et al. [26]	MCF-7	Potent cytostatic and cytotoxic effects
Ehigie et al. [27]	MDA-MB-231	Low mitochondrial membrane potential and intracellular ATP levels High ROS levels
Grossmann et al. [28]	MDA-MB-231 MDA-ERα	Cancer cell proliferation inhibition Apoptosis induction Loss of mitochondrial membrane potential Cell cycle arrest
Muhammad et al. [29]	MCF-7 MDA-MB-231	Apoptosis and autophagy induction In vivo tumor growth inhibition
Ray et al. [30]	MCF-7 MDA-MB-231	Cancer cell proliferation inhibition Amplified PARP cleavage Enhanced caspase activation
Nagasawa et al. [31]	SHN female mice	In vivo tumor growth inhibition with negligible side effects
Feng et al. [32]	4T1 MCF-7	Cancer cell proliferation inhibition ROS generation Mitochondrial function disruption In vivo tumor growth inhibition with negligible side effects
Shim et al. [33]	MDA-MB-231 MDA-MB-468	Low cholesterol and lipid expression levels In vivo tumor growth inhibition
Kilcar et al. [34]	MCF-7 MDA-MB-231	Significant estrogen receptor-dependent interaction after combination with paclitaxel

**Table 2 ijms-24-08918-t002:** The role of bitter melon in ovarian cancer prevention and therapy.

Study	Study Model	Main Findings
Martin et al. [35]	OVCAR4 OVCAR8 SKOV3 IGROV1	Apoptosis induction
Chan et al. [36]	A2780cp ES2 OVCA433 SKOV3 HOSE11-12 HOSE96-9-18 HEK293	Prevention of cancer cell migration, invasion, and proliferation Cell cycle arrest ROS generation Enhanced cytotoxic effects after combination with cisplatin In vivo tumor growth inhibition with negligible side effects
Pitchakarn et al. [37]	A2780 SKOV3	Enhanced cytotoxic effects after combination with cisplatin/paclitaxel
Yung et al. [38]	A2780cp A2780s C13 OV2008 SKOV3 OVCA433 ES2 HOSE17-1 HOSE96-9-18	Enhanced cytotoxic effects after combination with cisplatin In vivo tumor growth inhibition with negligible side effects

**Table 3 ijms-24-08918-t003:** The role of bitter melon in cervical cancer prevention and therapy.

Study	Study Model	Main Findings
Limtrakul et al. [39]	KB-V1 KB-3-1	Regulation of the function of P-glycoprotein and the multidrug resistance phenotype Sensitization to vinblastine
Pitchakarn et al. [40]	KB-V1 KB-3-1	Sensitization to vinblastine and paclitaxel
Pongnikorn et al. [41]	Cervical cancer patients undergoing radiotherapy	Negative impact on P-glycoprotein level on natural killer cell membrane

## Data Availability

Not applicable.

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
