# Peer review of "The Role of Bitter Melon in Breast and Gynecological Cancer Prevention and Therapy"

_ijms, 2023, doi:10.3390/ijms24108918_

Round 1

Reviewer 1 Report

This review addresses the anticancer effects of bitter melon against breast and gynecological cancers. Overall, the review is novel, timely, and has potential clinical significance. This reviewer has just a few minor comments, as follows.

1). Abstract: Different names of Momordica charantia appear in separate sentences without any explanation as to whether the same plant is mentioned throughout.

2). Section 3, p. 3 (top), and other Sections: “… we were able to identify a total of 19 relevant original research articles…” -  more publications related to the manuscript subject may appear during the peer review process. The article would benefit from incorporating these into a revised version.

3). Section 2, p. 2 (middle): “…Chinese and Indian lyophilized-juice and aqueous-methanolic extracts of Momordica charantia…” – This wording is misleading. The right one would be  “…lyophilized juice and aqueous-methanolic extract of Chinese and Indian Momordica charantia variants, respectively…” or a similar expression. There are several more incorrectly constructed sentences in the text. Thus, the entire manuscript needs to be proofread by an English editor to ensure correct grammar, phrasing, and terminology.

4). Section 9, p. 9 (top): “Of note, its bioactive components seem to interact with histone deacetylases and PPARs, which have been proposed to play a significant role in breast and gynecologic cancer development and progression[49-54]. – The authors inserted six self-citations (49-54) that do not directly relate to bitter melon. The number of such citations may be reduced.

5). Several references (1, 4, 8, 9) are incomplete. All references need to be checked and verified.

The manuscript needs to be proofread by an English editor to ensure correct grammar, phrasing, and terminology.

Author Response

This review addresses the anticancer effects of bitter melon against breast and gynecological cancers. Overall, the review is novel, timely, and has potential clinical significance. This reviewer has just a few minor comments, as follows.

1). Abstract: Different names of Momordica charantia appear in separate sentences without any explanation as to whether the same plant is mentioned throughout.

We have now replaced the different names with the unanimous term bitter melon in the abstract.

2). Section 3, p. 3 (top), and other Sections: “… we were able to identify a total of 19 relevant original research articles…” -  more publications related to the manuscript subject may appear during the peer review process. The article would benefit from incorporating these into a revised version.

Thank you for this useful remark. During the peer review process, no additional publications related to the manuscript subject appeared.

3). Section 2, p. 2 (middle): “…Chinese and Indian lyophilized-juice and aqueous-methanolic extracts of Momordica charantia…” – This wording is misleading. The right one would be  “…lyophilized juice and aqueous-methanolic extract of Chinese and Indian Momordica charantia variants, respectively…” or a similar expression. There are several more incorrectly constructed sentences in the text. Thus, the entire manuscript needs to be proofread by an English editor to ensure correct grammar, phrasing, and terminology.

We have now corrected the sentence, as suggested by the reviewer.

4). Section 9, p. 9 (top): “Of note, its bioactive components seem to interact with histone deacetylases and PPARs, which have been proposed to play a significant role in breast and gynecologic cancer development and progression[49-54]. – The authors inserted six self-citations (49-54) that do not directly relate to bitter melon. The number of such citations may be reduced.

The citations we inserted are relevant and directly refer to the statement that histone deacetylases and PPARs have been proposed to play a significant role in breast and gynecologic cancer development and progression.

5). Several references (1, 4, 8, 9) are incomplete. All references need to be checked and verified

We work with EndNote and have, indeed, faced this problem several times recently. So far, with the help of the MDPI office, we have always been able to complete these references during the proof-reading phase. Unfortunately, a correction at the current point is impossible, as EndNote would not synchronize with the changes. Thank you for your understanding and rest assured that we will take care of this issue during the proof-reading phase.

Reviewer 2 Report

This manuscript successfully summarizes the latest findings regarding the role of Bitter Melon in Breast and Gynecological Cancer Prevention and Therapy. Authors provide a detailed review of the latest reports employing breast and ovarian cancer cell lines and mice models. It is a well-written manuscript that worths to be published bearing in mind that there are only few reviews that summarize the biological effect of Momordica charantia against gynecological cancer types. Some minor changes must be conducted:

In order to be reader friendly, a graphical (schematic) representation must be included in the manuscript along with the generated tables.

The article "A comprehensive review on bitter gourd (Momordica charantia L.) as a gold mine of functional bioactive components for therapeutic foods" must be included in the Reference Section.

In Section 3. The Role of Bitter Melon in Breast Cancer Prevention and Therapy the phrase .. Human Epidermal growth factor Receptor 2 (HER2) must be re-written with a non-italic style, for uniformity purposes

Author Response

This manuscript successfully summarizes the latest findings regarding the role of Bitter Melon in Breast and Gynecological Cancer Prevention and Therapy. Authors provide a detailed review of the latest reports employing breast and ovarian cancer cell lines and mice models. It is a well-written manuscript that worths to be published bearing in mind that there are only few reviews that summarize the biological effect of Momordica charantia against gynecological cancer types. Some minor changes must be conducted:

In order to be reader friendly, a graphical (schematic) representation must be included in the manuscript along with the generated tables.

We have now added Figure 1 as a schematic representation along with the generated tables.

The article "A comprehensive review on bitter gourd (Momordica charantia L.) as a gold mine of functional bioactive components for therapeutic foods" must be included in the Reference Section.

We have now included the proposed article in the reference section.

In Section 3. The Role of Bitter Melon in Breast Cancer Prevention and Therapy the phrase ..Human Epidermal growth factor Receptor 2 (HER2) must be re-written with a non-italic style, for uniformity purposes

We have now rewritten the phrase with a non-italic style.